# Risk factors and pharmacotherapy for chemotherapy-induced peripheral neuropathy in paclitaxel-treated female cancer survivors: A retrospective study in Japan

**Shiori Hiramoto**[1,2], **Hajime Asano**[2], **Tomoyoshi Miyamoto**[1], **Manabu Takegami**[2], **Atsufumi Kawabata**[1] *

**1** Division of Pharmacology and Pathophysiology Faculty of Pharmacy, Kindai University, Higashi-Osaka, Japan, **2** Division of Pharmacy, Kindai University Hospital, Osakasayama, Japan

* kawabata@phar.kindai.ac.jp

**Data Availability Statement:** All relevant data are within the paper and supporting information files.

## Abstract

Chemotherapy-induced peripheral neuropathy (CIPN) is a dose-limiting adverse reaction in cancer patients treated with several cytotoxic anticancer agents including paclitaxel. Duloxetine, an antidepressant known as a serotonin-noradrenalin reuptake inhibitor, is the only agent that has moderate evidence for the use to treat painful CIPN. The present retrospective cohort study aimed to analyze risk factors for paclitaxel-induced peripheral neuropathy (PIPN), and investigate ongoing prescription drug use for PIPN in Japan. Female breast and gynecologic cancer patients who underwent paclitaxel-based chemotherapy at a single center in Japan between January 2016 and December 2019 were enrolled in this study. Patients' information obtained from electronic medical records were statistically analyzed to test possible risk factors on PIPN diagnosis. Patients' age, total paclitaxel dose, the history of female hormone-related diseases, hypertension and body mass index (BMI), but not additional platinum agents, were significantly associated with increased PIPN diagnosis. Drugs prescribed for PIPN included duloxetine, pregabalin, mecobalamin and Goshajinkigan, a polyherbal medicine, regardless of poor evidence for their effectiveness against CIPN, and were greatly different between breast and gynecologic cancer patients diagnosed with PIPN at the departments of Surgery and Gynecology, respectively. Thus, older age, greater total paclitaxel dose, the history of estrogen-related diseases, hypertension and BMI are considered risk factors for PIPN in paclitaxel-based chemotherapy of female cancer patients. It appears an urgent need to establish a guideline of evidence-based pharmacotherapy for PIPN.

## Introduction

Cancer chemotherapy with anti-cancer drugs, such as taxanes, platinum-containing agents, vinca alkaloids and proteasome-inhibiting agents including bortezomib, often causes

**Funding:** The authors received no specific funding for this work.

**Competing interests:** The authors have declared that no competing interests exist.

chemotherapy-induced peripheral neuropathy (CIPN), a potentially dose-limiting adverse reaction, which impairs patients' quality of life [1–4]. The pathogenesis of CIPN still remains largely unclear, although preclinical studies have suggested possible involvement of oxidative stress, various humoral factors, and altered function or expression of ion channels and receptors in the development or maintenance of CIPN [5]. The detailed molecular mechanisms underlying CIPN appear to vary with types of chemotherapeutic agents, e.g. resident and infiltrating macrophages play a critical role in the CIPN following administration of paclitaxel, a microtubule-stabilizing agent, but not oxaliplatin, a platinum agent [6–9]. Clinically, paclitaxel is widely used to treat breast and gynecologic (ovarian, fallopian tube, endometrial) cancers in females, leading to CIPN with high frequency [4, 10]. Among platinum agents capable of causing CIPN [4, 11], carboplatin is often included in the paclitaxel-based, front-line regimens, whereas the incidence of CIPN following paclitaxel and carboplatin combination treatment is similar to or less than that following paclitaxel alone [12–15]. Our clinical and preclinical studies have shown that 57 years of age or older and endocrine therapy were significantly associated with severer PIPN at a private hospital in Izumi city, Japan, and that ovariectomy aggravated PIPN in laboratory animals, an effect reversed by estrogen supplementation [16].

According to the updated American Society of Clinical Oncology (ASCO) guidelines, no agents are available for prevention of CIPN, and duloxetine is the only agent that has appropriate evidence to support its use for treatment of established painful CIPN [17], although its effectiveness on taxane-induced painful CIPN is controversial [18]. The moderate usefulness of duloxetine for treatment of CIPN in cancer patients treated with various cytotoxic chemotherapeutics including paclitaxel has also been supported by clinical trials in Japan [19, 20]. Nonetheless, apart from opioids and non-steroidal anti-inflammatory drugs (NSAIDs) that are often used to treat cancer pain, vitamin B12, pregabalin and Goshajinkigan, a Chinese polyherbal medicine, in addition to duloxetine, are commonly prescribed for treatment of CIPN in Japan [21], although there is poor evidence for their effectiveness against CIPN [17, 22–24].

In the present study, we retrospectively analyzed the association of various factors including older age, cumulative dose of paclitaxel and addition of platinum agents with PIPN diagnosis in female breast or gynecologic cancer survivors who underwent paclitaxel-based chemotherapy at Kindai University Hospital in Osakasayama city, Japan, and examined current trends in prescription drug use to treat PIPN.

## Methods

### Study design, setting and patient data collection

This was a single-center, retrospective cohort study using data obtained from electronic medical records of female patients who were diagnosed with breast cancer or gynecologic (ovarian, fallopian tube or endometrial) cancer and underwent paclitaxel-based chemotherapy at Kindai University Hospital in Osakasayama City, Japan, between 1st January, 2016, and 31st December, 2019. The collected patient information included age, body surface area (BSA), body mass index (BMI), metastasis at initial diagnosis, treatment history, ongoing diseases, such as diabetes, hyperlipidemia and hypertension, medical history of non-cancer, female hormone-related diseases (defined as diseases the pathogenesis of which involves altered levels of estrogens) including ovarian cyst, uterine fibroid and endometriosis, typical diseases associated with estrogens [25–27]. Information concerning the diagnosis of PIPN was also collected retrospectively from the medical record. At the departments of Surgery (breast cancer) and Gynecology (gynecologic cancers), Kindai University Hospital, PIPN was routinely diagnosed by physicians or by nurses or pharmacists under a physician's direction according to CTCAE version 5.0. In treatment history, we also checked prescription of duloxetine, pregabalin, mecobalamine (vitamin B12) and

Goshajinkigan, a polyherbal medicine, which are often used for treatment of painful peripheral neuropathy including CIPN in Japan, after the onset of paclitaxel treatment.

## Ethics approval

This study was approved by Ethics Committees of Kindai University, Faculty of Medicine (approval number 31–222, January 21, 2020) and Faculty of Pharmacy (approval number 20–156, April 22, 2020), which waved the requirement of informed consent, owing to the retrospective nature of this study and approved the use of an opt-out strategy concerning patient consent. Namely, patients were included in the research unless they requested to be excluded, and our clinical study information was communicated to the patients through the institutional website. All data/samples were fully anonymized before the authors accessed them, and any members of our research team named in the author list did not have direct access to patient identification when analyzing the data from patients' information.

## Inclusion and exclusion criteria

Inclusion criteria were two or more administrations of paclitaxel, no history of paclitaxel treatment prior to the observation period, no participation in a clinical trial, no prior use of oxaliplatin (by far the highly neurotoxic platinum agent and not used for treatment of breast or gynecologic cancers) or vinca alkaloids, no use of nanoparticle albumin-bound (nab)-paclitaxel, no preexisting neuropathy-like symptoms including numbness, no prescription of duloxetine, pregabalin, mecobalamine or Goshajinkigan before the onset of paclitaxel treatment. Of 431 female patients with beast or gynecological cancer who underwent paclitaxel-based chemotherapy, 148 who did not meet the inclusion criteria were excluded and the residual 283 were enrolled in this study.

## Statistical analysis

The total dose of paclitaxel and age are shown using box-and-whisker plots, and their differences between breast cancer patients and gynecologic cancer patients were statistically analyzed by Mann-Whitney's U test. The association of various factors with PIPN diagnosis was analyzing by Fisher's exact test in a dichotomous manner. Continuous variables were divided into two categories using median splits. Age and total dose of paclitaxel were also categorized into two groups using optimal cutoff values determined from the receiver operating characteristic (ROC) curves. Multivariate logistic regression analysis was conducted to test the association of age, total dose of paclitaxel and addition of platinum agents with PIPN diagnosis, showing odds ratios with 95% confidence intervals (CIs). Kaplan-Meier curves were generated to visualize the time-related increase in PIPN diagnosis in two separate groups, and their differences between the two were analyzed by the log-rank test. Hazard ratios with 95% CIs were also calculated using a Cox proportional hazard regression model. A $p$ value less than 0.05 was considered statistically significant. We used EZR (Saitama Medical Center, Jichi Medical University, Saitama, Japan), a graphical user interface for R (version 3.6.3, R Foundation for Statistical Computing, Vienna, Austria), which was a modified version of R commander (version 2.6–2), as described elsewhere [28].

## Results

### Characteristics of patients enrolled in the present study

In this study, 283 patients including 162 breast and 121 gynecologic cancer survivors who met inclusion criteria (S1 Fig) were enrolled for statistical analysis (Table 1). Of the paclitaxel-

**Table 1. Clinical characteristics of female cancer patients enrolled in this study.**

| | n | n/n$_{Total}$ (%) |
|---|---|---|
| Cancer type | | |
| Breast cancer | 162 | 57.2 |
| Gynecologic cancer | 121 | 42.8 |
| Addition of platinum agents | | |
| No | 162 | 57.2 |
| Yes | 121 | 42.8 |
| Diagnosis of PIPN | | |
| No | 45 | 15.9 |
| Yes | 238 | 84.1 |

The total number of the enrolled patients was 283.

treated breast cancer survivors, 138 and 24 received weekly paclitaxel and dose-dense pacli-
taxel, respectively, while 115 of the paclitaxel-treated gynecologic cancer patients received tri-
weekly paclitaxel (S1 Table). It is to be noted that nobody of the 162 paclitaxel-treated breast
cancer patients received additional administration of platinum agents, while all of 121 pacli-
taxel-treated gynecologic cancer patients received platinum agents including carboplatin
(n = 120) and cisplatin (n = 1) (Table 1; S1 Table). The total dose of paclitaxel was significantly
greater in gynecologic cancer patients than in breast cancer patients (S2 Fig). There was no sig-
nificant difference of age between breast and gynecologic cancer patients (S2 Fig).

## Association of various factors with diagnosis of PIPN in female cancer survivors who underwent paclitaxel-based chemotherapy

Fisher's exact test indicated significant association of Age $\geq$ 58 and total paclitaxel dose $\geq$ 944.9
(medians), but not cancer types or additional administration of platinum agents, with PIPN
diagnosis in female cancer survivors, which was supported by multivariate logistic regression
analysis [Odds ratios for age and total paclitaxel dose over the medians, 2.03 (95% CI, 1.03–4.00;
$p$ = 0.040) and 2.83 (95% CI, 1.35–5.91; $p$ = 0.00575), respectively] (Table 2). The optimal cutoff
values of age and total dose of paclitaxel in analysis of their association with PIPN diagnosis were
55 years of age and 924.8 mg/m$^2$, respectively, as estimated by ROC analysis (Table 2). Interest-
ingly, medical history of female hormone-related diseases, hypertension and BMI $\geq$ 21.8
(median), but not surgery, metastasis, diabetes hyperlipidemia or BSA $\geq$ 1.526 m$^2$ (median)
were significantly associated with PIPN diagnosis, as evaluated by Fisher's exact test (Table 2).

Kaplan-Meier curves show time-related increase in PIPN diagnosis (Fig 1). The log-rank
test indicated significant impact of age $\geq$ 58 (median) and $\geq$ 55 (optimal cut-off value) years
on PIPN diagnosis ($p$ = 0.0103 and 0.0153, respectively), and the Cox proportional hazard
model provided hazard ratios, 1.398 (95%CI, 1.082–1.805) and 1.383 (95%CI, 1.064–1.799),
respectively (Fig 2A). On the other hand, total paclitaxel dose, regardless using medians or
optimal cut-off values for categorization, or additional platinum agents had no significant
impact on PIPN diagnosis (Fig 2B and 2C).

## Pharmacotherapy to treat PIPN in female cancer survivors who underwent paclitaxel-based chemotherapy

Of 283 enrolled patients, 124 (43.8%) received prescription of one or more of duloxetine, preg-
abalin, mecobalamin and Goshajinkigan (Table 3), well-known medicines used for treatment

**Table 2. Association of various factors with diagnosis of PIPN in female cancer patients.**

| | Diagnosis of PIPN | | | | |
| | % | | | Multivariate | |
| Factor | (Yes/Yes+No) | *p* | Odds ratio (95% CI) | *p* |
|---|---|---|---|---|
| Cancer type | | | | |
| Breast cancer | 84.6 (137/162) | | | |
| Gynecologic cancer | 83.5 (101/121) | 0.870 | | |
| Age (Median: 58) | | | | |
| < 58 | 79.6 (109/137) | | Reference | |
| ≥ 58 | 88.4 (129/146) | 0.0511 | 2.03 (1.03–4.00) | **0.040** |
| Age (Optimal cutoff value: 55) | | | | |
| < 55 | 77.8 (91/117) | | | |
| ≥ 55 | 88.6 (147/166) | **0.0202** | | |
| Total dose of PCT (Median: 944.9) | | | | |
| < 944.9 mg/m$^2$ | 78.0 (110/141) | | Reference | |
| ≥ 944.9 mg/m$^2$ | 90.1 (128/142) | **0.00575** | 2.83 (1.35–5.91) | **0.00575** |
| Total dose of PCT (Optimal cutoff value: 924.8) | | | | |
| < 924.8 mg/m$^2$ | 74.6 (85/114) | | | |
| ≥ 924.8 mg/m$^2$ | 90.5 (153/169) | **0.000439** | | |
| Addition of platinum agents | | | | |
| No | 84.6 (137/162) | | Reference | |
| Yes | 83.5 (101/121) | 0.870 | 0.669 (0.331–1.35) | 0.261 |
| Surgery | | | | |
| No | 86.7 (13/15) | | | |
| Yes | 84.0 (225/268) | 1.00 | | |
| Metastasis | | | | |
| No | 86.7 (117/135) | | | |
| Yes | 81.8 (121/148) | 0.329 | | |
| Medical history of female hormone-related diseases | | | | |
| No | 82.0 (205/250) | | | |
| Yes | 100 (33/33) | **0.00403** | | |
| Diabetes | | | | |
| No | 83.3 (224/269) | | | |
| Yes | 100 (14/14) | 0.137 | | |
| Hyperlipidemia | | | | |
| No | 84.0 (226/269) | | | |
| Yes | 85.7 (12/14) | 1.00 | | |
| Hypertension | | | | |
| No | 81.8 (189/231) | | | |
| Yes | 94.2 (49/52) | **0.0338** | | |
| BSA (Median: 1.526) | | | | |
| < 1.526 | 82.3 (116/141) | | | |
| ≥ 1.526 | 85.9 (122/142) | 0.421 | | |
| BMI (Median: 21.8) | | | | |
| < 21.8 | 78.2 (111/142) | | | |
| ≥ 21.8 | 90.1 (127/141) | **0.00878** | | |

The association between various factors and PIPN diagnosis was analyzed in a dichotomous manner, where continuous variables were divided into two categories using medians. The optimal cutoff values for patients' age and total dose of paclitaxel (PCT) were also determined from the receiver operating characteristic (ROC) curves, and used for categorization. BSA, body surface area; BMI, body mass index. Effect of each factor on PIPN diagnosis was analyzed by Fisher's exact test. Further, effects of age, total dose of PCT and addition of platinum agents on PIPN diagnosis were evaluated by multivariate logistic regression analysis. CI, confidence interval.

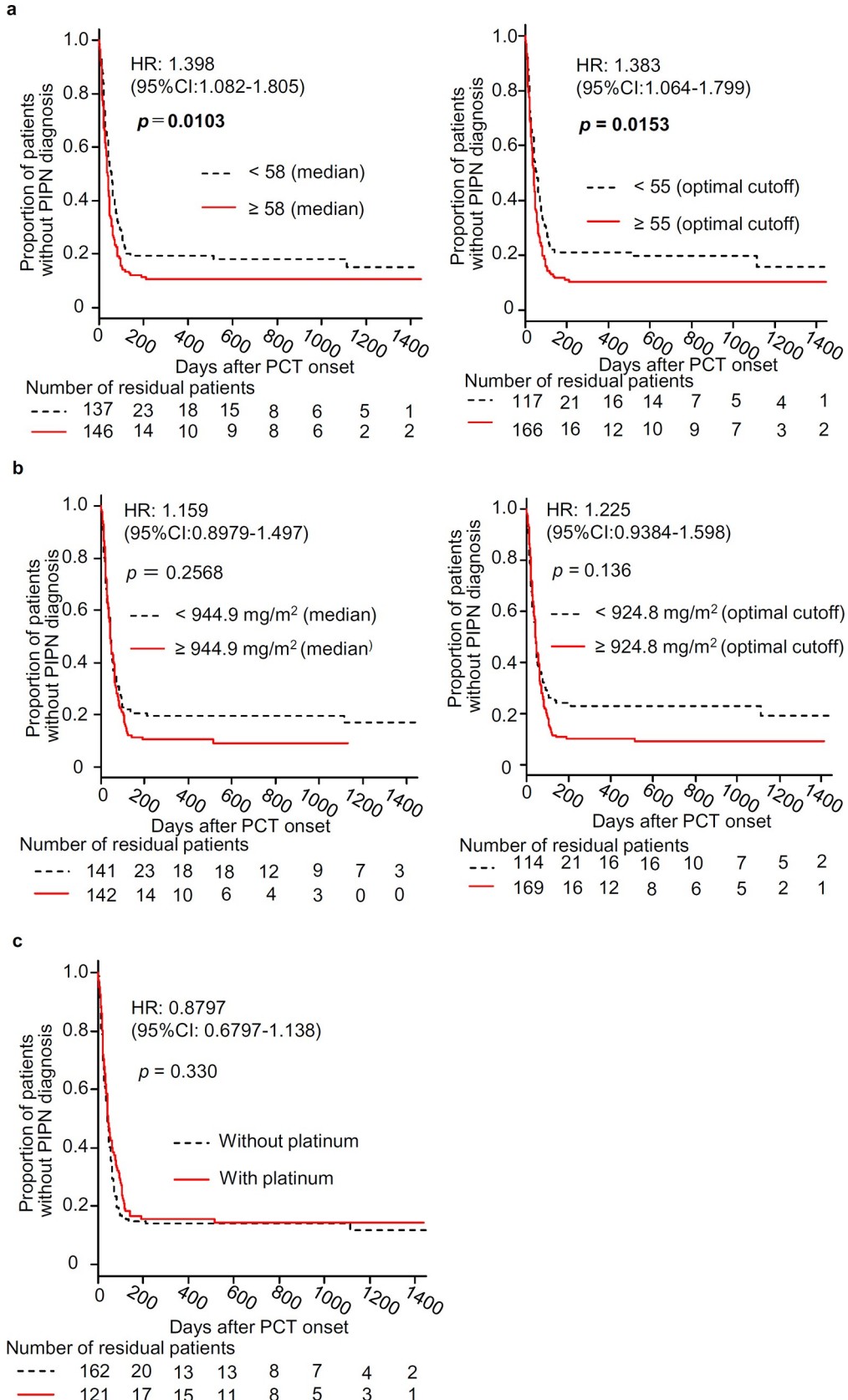

**Fig 1. Kaplan-Meier curves for the time-related increase in PIPN diagnosis in female cancer patients treated with paclitaxel (PCT), and their association with older age (a), greater total paclitaxel dose (b) and additional administration of platinum agents (c).** Medians were 58 years of age (a, left) and 944.9 mg/m$^2$ of total paclitaxel dose (b, left), and optimal cutoff values estimated by ROC analysis were 55 (a, right) and 924.8 (b, right), respectively. Statistical significance between two groups was analyzed by a log-rank test. Hazard ratios (HRs) with 95% CIs were calculated using a Cox proportional hazard regression model.

of peripheral neuropathy in Japan [21]. It is noteworthy that 4 patients received duloxetine, pregabalin or mecobalamin without PIPN diagnosis (Table 3). Of 238 cancer patients with PIPN diagnosis (see Table 1), 20–30% received pregabalin, mecobalamin or Goshajinkigan, while only 14% received duloxetine (Fig 2). Interestingly, the prescription rates of duloxetine, mecobalamin and Goshajinkigan were greater in gynecologic cancer patients diagnosed with PIPN at Gynecology department than in breast cancer patients diagnosed with PIPN at Surgery department, while the prescription rate of pregabalin was rather greater in the latter patients than in the former (Fig 2).

## Discussion

Our retrospective cohort study showed that older age and increased total paclitaxel dose were associated with increases in PIPN diagnosis in female breast and gynecologic cancer patients, in agreement with the previous reports [16, 29–31]. Our study also identified the significant impact of the medical history of female hormone-related diseases, hypertension and BMI on PIPN development. On the other hand, additional administration of platinum agents, such as carboplatin and cisplatin (120 and 1 cases, respectively) was not associated with increased PIPN diagnosis, as reported previously [12–16].

Our recent preclinical study demonstrated that ovariectomy dramatically aggravated the development of PIPN in an estrogen-reversible manner in mice, suggesting that estrogen protects against PIPN development [16] and supporting clinical evidence for the impact of older age on the increased incidence or severity of PIPN in the present and previous studies [16, 29–31]. Thus,

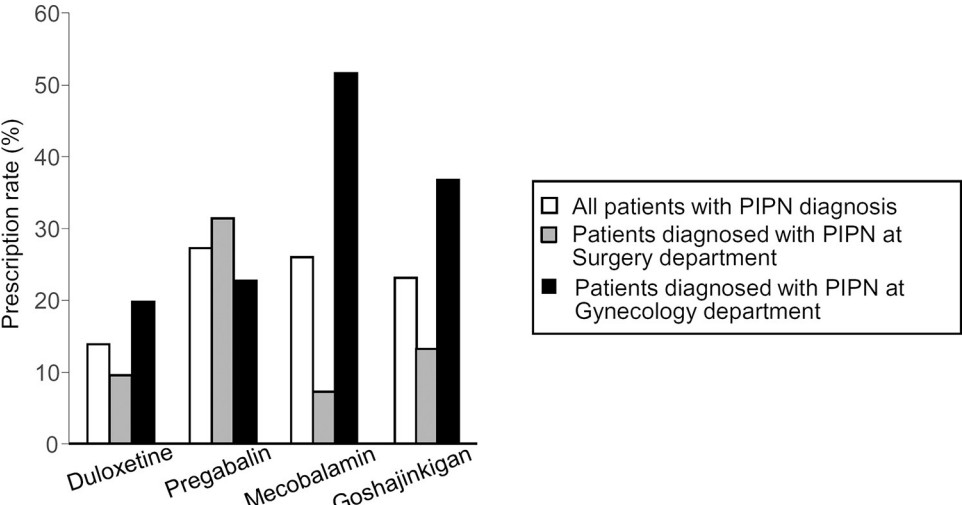

**Fig 2. Prescription drug use for PIPN in female cancer patients who were diagnosed with PIPN at the departments of Surgery and Gynecology.** Of 283 enrolled female cancer patients, 238 including 137 breast and 101 gynecologic cancer patients were diagnosed with PIPN at the departments of Surgery and Gynecology, respectively. Data show the prescription rate of each drug among patients diagnosed with PIPN.

**Table 3. Pharmacotherapy to treat PIPN in female cancer patients.**

| Medication | n | PIPN | |
| --- | --- | --- | --- |
| | | Yes | No |
| Duloxetine | 34 | 33 | 1 |
| Pregabalin | 66 | 65 | 1 |
| Mecobalamin | 64 | 62 | 2 |
| Gosyajinkigan | 55 | 55 | 0 |

Of 283 enrolled female cancer patients, 124 received prescription of one or more of duloxetine, pregabalin, mecobalamin and Goshajinkigan for treatment of PIPN.

the present study ascertains that the estrogen decline, particularly in postmenopausal cancer patients, is a risk factor of PIPN. Taken together, the present results indicating the association of the history of estrogen-related diseases including ovarian cyst, uterine endometriosis and leiomyomas with PIPN development might suggest the critical role of estrogens in the pathophysiology of PIPN. Particularly, it has been reported that endometriosis induces somatic pain or allodynia/hyperalgesia containing inflammatory and neuropathic components, in addition to visceral pain [32, 33]. Further clinical and preclinical studies are necessary to clarify the molecular mechanisms underlying the involvement of the altered estrogen levels in PIPN.

The present study identified significant association of hypertension and increased BMI with PIPN diagnosis (see Table 2). Interestingly, there is clinical evidence for the association between hypertension and CIPN in cisplatin-treated patients [34]. Whether hypertension itself or pharmacotherapy for hypertension is associated with CIPN/PIPN is still open to question.

It is to be noted that all of paclitaxel-treated gynecologic cancer patients received platinum agents, while nobody of paclitaxel-treated breast cancer patients was treated with platinum agents in the present study, thereby providing a potential bias. It is also noteworthy that the total paclitaxel dose was slightly but significantly greater in gynecologic cancer patients than in breast cancer patients (see S2 Fig).

As described in the previous report from survey of the management of CIPN in Japan [21], the present study ascertained that many of female cancer patients with PIPN were prescribed with pregabalin, mecobalamin or Goshajinkigan (see Table 3) in spite of poor evidence for their effectiveness against CIPN [17, 22–24], while relatively few patients with PIPN received duloxetine, the only medicine that has moderate evidence for the use to treat the established CIPN [17] (see Table 3 and Fig 2). Goshajinkigan, a polyherbal medicine that legally requires a prescription to be dispensed in Japan, contains fixed proportions of 10 crude herbal extracts, i.e. achyranthes root, rehmannia root, dioscorea, rhizome, cornus fruit, alisma rhizome, plantago seed moutan bark, pariah sclerotium, aconite root and cinnamon bark [35]. Goshajinkigan is widely used for treatment of diabetic neuropathy, and appears to have limited effectiveness on CIPN [22, 35, 36]. Surprisingly, there were great differences in drugs prescribed to treat PIPN at the departments of Surgery for breast cancer patients and of Gynecology for gynecologic cancer patients. This might simply reflect different preference of physicians belonging to different departments, but is still open to question. Pharmacists may be able to contribute to the improvement of treatment of CIPN/PIPN in female cancer survivors undergoing chemotherapy through identifying high-risk patients and making appropriate evidence-based recommendations.

There are several limitations of this study. This is a single-center retrospective study that had a relatively small sample size. As did all other observational studies, the present study possibly had unmeasured confounding factors, which could lead to biased effect estimates. In

addition, as mentioned above, no breast cancer patients had additional platinum agents, while all gynecologic cancer patients received administration of carboplatin or cisplatin. Therefore, we could not exactly discriminate the effects of additional platinum agents and differences of cancer types or medical care departments on PIPN incidence. Despite these limitations, we argue that this study is useful to ascertain risk factors for the development and progression of PIPN and to highlight currently ongoing prescription drug use for PIPN in Japan.

## Conclusions

In paclitaxel-treated female breast and gynecologic cancer patients, older age and increased total paclitaxel dose were significantly associated with the incidence of PIPN, and the history of female hormone-related disease, hypertension and BMI also had significant impact on PIPN development. On the other hand, additional administration of platinum agents had no significant impact on PIPN diagnosis. Our study also highlighted the current trends of pre-scription drugs used for treatment of PIPN, which included duloxetine, pregabalin, mecobala-min and Goshajinkigan, regardless of the limited scientific evidence for their effectiveness against CIPN. Together, older age, greater total paclitaxel dose, medical history of female hor-mone-related diseases, hypertension and BMI should be considered risk factors for PIPN development in paclitaxel-based chemotherapy of female cancer patients. Thus, there is an urgent need to establish a guideline of evidence-based pharmacotherapy for PIPN available for cancer patients undergoing chemotherapy at different departments.

## Supporting information

**S1 Fig. Diagram of patient selection.** PCT, paclitaxel. *Two patients had both ovarian and fal-lopian tube cancers.
(PDF)

**S2 Fig. Comparison of total dose of paclitaxel (PCT) and age between breast and gyneco-logic cancer patients.** The top graphs show box-and-whisker plots. Statistical significance was analyzed by Mann-Whitney's U test.
(PDF)

**S1 File.**
(PDF)

**S1 Table. Chemotherapy regimens used in female breast and gynecologic cancer patients treated with paclitaxel at Kindai University Hospital.** PCT, paclitaxel; ddPCT, dose-dense paclitaxel; wPCT, weekly paclitaxel; HP, trastuzumab + pertuzumab; BEV, bevacizumab; HER, trastuzumab; ddTC, dose-dense paclitaxel + carboplatin; TP, paclitaxel + cisplatin; tri-weekly TC, tri-weekly paclitaxel + carboplatin; wTC, weekly paclitaxel + carboplatin.
(PDF)

## Author Contributions

**Conceptualization:** Atsufumi Kawabata.

**Data curation:** Shiori Hiramoto, Hajime Asano.

**Formal analysis:** Shiori Hiramoto, Tomoyoshi Miyamoto.

**Investigation:** Shiori Hiramoto, Hajime Asano, Manabu Takegami.

**Supervision:** Hajime Asano, Manabu Takegami, Atsufumi Kawabata.

**Writing – original draft:** Shiori Hiramoto, Atsufumi Kawabata.

**Writing – review & editing:** Shiori Hiramoto, Tomoyoshi Miyamoto, Atsufumi Kawabata.

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
