## [Decision Letter · Decision Letter 0]

21 Oct 2021

PONE-D-21-29568Association between risk factors and pharmacotherapy of chemotherapy-induced peripheral neuropathy in paclitaxel-treated female cancer survivors: A retrospective study in JapanPLOS ONE

Dear Dr. Kawabata,

Thank you for submitting your manuscript to PLOS ONE. After careful consideration, we feel that it has merit but does not fully meet PLOS ONE’s publication criteria as it currently stands. Therefore, we invite you to submit a revised version of the manuscript that addresses the points raised during the review process.

We look forward to receiving your revised manuscript.

Kind regards,

Nial J. Wheate, Ph.D.

Academic Editor

PLOS ONE

Journal Requirements:

2. In ethics statement in the manuscript and in the online submission form, please provide additional information about the patient records/samples used in your retrospective study. Specifically, please ensure that you have discussed whether all data/samples were fully anonymized before you accessed them.

Additional Editor Comments:

Please revise and resubmit the article based on the comments of both reviewers.

Reviewers' comments:

Reviewer's Responses to Questions

**Comments to the Author**

1. Is the manuscript technically sound, and do the data support the conclusions?

Reviewer #1: Partly

Reviewer #2: Yes

2. Has the statistical analysis been performed appropriately and rigorously? 

Reviewer #1: Yes

Reviewer #2: Yes

3. Have the authors made all data underlying the findings in their manuscript fully available?

Reviewer #1: Yes

Reviewer #2: No

4. Is the manuscript presented in an intelligible fashion and written in standard English?

Reviewer #1: Yes

Reviewer #2: Yes

5. Review Comments to the Author

Reviewer #1: Hiramoto et al. conducted a retrospective chart review to identify predictors of paclitaxel-induced peripheral neuropathy. They find some associations that are pretty well established including age and paclitaxel dose. They also find some interesting findings, particularly the hormone-related diseases association that connects to their prior work on estrogen levels. The major concern is that several of the analyses with the downstream endpoints of duloxetine or other neuropathy-agent use should not be conducted because they are highly confounded with the primary analysis of PIPN. Pending resolution of the following suggestions this manuscript could be acceptable for publication.

Major:

-The methods must explain how “diagnosis of PIPN” was defined. Who diagnosed this? Was it based on clinician prospective evaluation or retrospective collection from the medical record? Were PIPN assessments mandated? Was

a patient-reported questionnaire used? Etc.

-The association of clinical factors with prescription of duloxetine or other agents for PIPN diagnosis is not worth analyzing or reporting. The decision to prescribe these agents is substantially confounded by the PIPN diagnosis, which was already directly analyzed for association with these clinical variables. These entire analyses should be removed. Instead, you should just report the descriptive statistics of which drugs were prescribed for PIPN (currently in table 1) and perhaps analyze the association of PIPN diagnosis (however defined) with PIPN treatment. The abstract needs to be rewritten after removing these unnecessary analyses.

-The “time-related increases” section could be merged with the previous sections as this is just an alternative analysis method to answer the same questions. Again, this should not be done for duloxetine and other agents since these are indirect associations mediated by PIPN. Just analyze the association with PIPN and merge into the previous section.

Minor:

-Line 48: duloxetine is only for painful CIPN, not sensory or motor. It was also effective only for platinum-induced, not taxane-induced, painful CIPN in the JAMA trial (Smith et al., https://pubmed.ncbi.nlm.nih.gov/23549581/) (albeit in an unplanned secondary analysis). This should be corrected throughout, including the last sentence of the conclusion (which should not definitively recommend duloxetine for all PIPN) and abstract.

-Line 94: why did you exclude oxaliplatin and then conduct analyses of the other platinums?

-Line 109: The optimal cutoff method is very problematic as it allows for any possible association to be detected. Why was this used instead of median? Or a continuous variable using logistic regression?

-Figure 1: The objective of this study is not to explore differences between breast and gynecologic cancer in terms of age or paclitaxel dose. This is not a sufficiently informative figure to include in the main manuscript. It can be moved to supplement.

-Table 2: Please replace columns 1, 2, and 3 with one column that reports the % (#s) for yes (for example, row 1 would read: breast cancer, 84.6% (136/162)).

-Line 72: Please further define “female hormone-related diseases” since this is an important variable in your analysis. Was there a complete list of diseases that would have qualified? Or that did qualify?

-Line 282: You do not have any data on CIPN severity and cannot conclude that platinums enhance severity and not incidence from your data. Also delete lines 342-344 for the same reason and because this analysis should be removed.

-Lines 284-290: It would be worthwhile to mention that “female hormone-related diseases” were associated with PIPN within this paragraph since it also fits with this hypothesis (currently lines 319-322). I think more should be made of this finding in the context of your work.

-Lines 314-319: It is confusing that the discussion goes through risk factors, then treatment strategies, and then goes back to risk factors. Please move this content up to before the treatment strategies paragraph.

Optional/typographical:

-Table 1: Replace “operation” with “surgery”

Reviewer #2: Overall a well designed, conducted, and interesting article. I offer the following comments before it is accepted for publication:

1. I'm not familiar with the herbal medicine goshajinkigan. The article looks at patients prescribed the medicine, but is it possible for patients to be taking it without a prescription (e.g. over-the-counter, given that is herbal). If so, how do you know other patients were not taking it which would skew the results?

2. Inclusion criteria (line 94) says: no dministration of oxaliplatin or vinca alkaloids. Presumably this means "no prior use of" the drugs as there are patients included it the analysis who were given a platinum. Suggest slight rewording to make it clear.

3. Line 128. The study included only 283 patients, so why does the data in the results (lines 128-129) give numbers for patients also excluded from the study. How is that information relevant? Suggest this is changed so only discusses patients included in the study

6. PLOS authors have the option to publish the peer review history of their article (what does this mean?). If published, this will include your full peer review and any attached files.

Reviewer #1: **Yes: **Dan Hertz

Reviewer #2: No

---

## [Author Response · Author response to Decision Letter 0]

30 Nov 2021

Responses to Reviewer #1

We thank the reviewer #1 for understanding significance of our study and providing important opinions useful to improve our manuscript. 

R1-1) The methods must explain how “diagnosis of PIPN” was defined. Who diagnosed this? Was it based on clinician prospective evaluation or retrospective collection from the medical record? Were PIPN assessments mandated? Was a patient-reported questionnaire used? Etc.

This is a very important opinion that we should consider. Information concerning the diagnosis of PIPN was collected retrospectively from the medical record. At the departments of Surgery (breast cancer patients) and Gynecology (gynecologic cancers patients, Kindai University Hospital, PIPN was routinely diagnosed by physicians or by nurses or pharmacists under a physician’s direction according to CTCAE version 5.0. These comments have been described in the ‘Methods’ section (Lines 75-79).

R1-2) The association of clinical factors with prescription of duloxetine or other agents for PIPN diagnosis is not worth analyzing or reporting. The decision to prescribe these agents is substantially confounded by the PIPN diagnosis, which was already directly analyzed for association with these clinical variables. These entire analyses should be removed. Instead, you should just report the descriptive statistics of which drugs were prescribed for PIPN (currently in table 1) and perhaps analyze the association of PIPN diagnosis (however defined) with PIPN treatment. The abstract needs to be rewritten after removing these unnecessary analyses.

We thank Reviewer #1 for providing kind advices. To address the reviewer’s requests, we have made a lot of revisions in the text, tables and figures, as follows.

R1-2-1: According to the reviewer’s suggestions, original Tables 3 and 4 showing the association of clinical factors with prescription of duloxetine or other agents for PIPN have been withdrawn, and the related sentences in the text including Abstract have been deleted (Lines 11-14, 18-21, 56-60, 115, 120, 122, 141, 165-179, 197-209, 227, 228-230, 232-233, 237, 248, 251, 254, 256, 282, 289-293, 296-299). 

R1-2-2: The description concerning ‘Drugs prescribed to treat PIPN’ in original Table 1 has been deleted. Alternatively, descriptive statistics of drugs for PIPN in association with PIPN diagnosis have been shown in new Table 3 and new Fig. 2. The related explanations and modifications have been made in the text including Abstract (Lines 14-18, 20-21, 60, 144-150, 197-221, 259, 262-271, 293-296, 299-301).

R1-2-3: Considering the above revisions in response to the reviewer, the title of our manuscript has been altered to “Risk factors and pharmacotherapy for chemotherapy-induced peripheral neuropathy in paclitaxel-treated female cancer survivors: A retrospective study in Japan” (Page 1).

R1-3) The “time-related increases” section could be merged with the previous sections as this is just an alternative analysis method to answer the same questions. Again, this should not be done for duloxetine and other agents since these are indirect associations mediated by PIPN. Just analyze the association with PIPN and merge into the previous section.

As suggested by Reviewer #1, the “time-related increases” section has been merged with the “Association of various factors with diagnosis of PIPN in female cancer survivors who underwent paclitaxel-based chemotherapy” section. The data of the analysis of the time-related correlations between prescription drug use for PIPN treatment and factors including age, paclitaxel dose or platinum addition have been deleted from original Fig 2 (now, new Fig 1). The related parts of the text have been modified (Lines 179-194).

R1-4) Line 48: duloxetine is only for painful CIPN, not sensory or motor. It was also effective only for platinum-induced, not taxane-induced, painful CIPN in the JAMA trial (Smith et al., https://pubmed.ncbi.nlm.nih.gov/23549581/) (albeit in an unplanned secondary analysis). This should be corrected throughout, including the last sentence of the conclusion (which should not definitively recommend duloxetine for all PIPN) and abstract.

I thank Reviewer #1 for suggesting an important point. The JAMA trial paper by Smith et al. has been cited and mentioned in Introduction (Lines 47-48), and the sentences indicating definitive recommendation of duloxetine for PIPN treatment in the text including Abstract and Conclusions have been modified (Lines 5, 20-21, 47-50,293-296, 299-301). 

R1-5) Line 94: why did you exclude oxaliplatin and then conduct analyses of the other platinums?

The original sentence “-----no administration of oxaliplatin-----” was not clear, and altered to “-----no prior use of oxaliplatin” (Lines 101-103). In the present study, a few subjects had the history of the use of oxaliplatin for treatment of cancers other than breast and gynecologic cancers. Among different platinum agents, oxaliplatin most frequently causes severe CIPN. We thus considered that the prior use of oxaliplatin might affect the development of CIPN in breast and gynecologic cancer patients treated with paclitaxel and/or the other platinum agents including carboplatin. We then decided to exclude patients who had the history of oxaliplatin use. Considering the reviewer’s comment, we incorporated additional brief explanation into the text (Lines 101-103). 

R1-6) Line 109: The optimal cutoff method is very problematic as it allows for any possible association to be detected. Why was this used instead of median? Or a continuous variable using logistic regression?

I thank Reviewer #1 for giving an opinion. Considering the reviewer’s opinion, we used a median split, in addition to the optimal cut-off values, for categorization in new Table 2 and new Fig 1, though the cut-off values determined by ROC analysis are useful when certain values other than medians have specific meanings (e.g. menopausal age in this study). The related parts of the text have been modified (Lines 116-118, 155-162, 168-177, 179-185, 187-194).

R1-7) Figure 1: The objective of this study is not to explore differences between breast and gynecologic cancer in terms of age or paclitaxel dose. This is not a sufficiently informative figure to include in the main manuscript. It can be moved to supplement.

As suggested by Reviewer #1, original Fig 1 has been altered to a supplementary figure, S2 Fig. The related parts of the text have been modified (Lines 142-144).

R1-8) Table 2: Please replace columns 1, 2, and 3 with one column that reports the % (#s) for yes (for example, row 1 would read: breast cancer, 84.6% (136/162)).

New Table 2 has been modified as suggested by Reviewer #1 (Lines 168-177).

R1-9) Line 72: Please further define “female hormone-related diseases” since this is an important variable in your analysis. Was there a complete list of diseases that would have qualified? Or that did qualify?

In the present study, “female hormone-related diseases” were defined as diseases the pathogenesis of which involves altered levels of estrogens, and included ovarian cyst, uterine endometriosis and leiomyomas, typical diseases associated with estrogens (Int J Mol Sci. 2020, 21 (8), 2815; Biomec Res Int. 2021, 2021, 6660087; Reprod Sci. 2016, 23 (2), 163-175). This has been explained in the text (Lines 73-75).

R1-10) Line 282: You do not have any data on CIPN severity and cannot conclude that platinums enhance severity and not incidence from your data. Also delete lines 342-344 for the same reason and because this analysis should be removed.

Considering Reviewer #1’s opinions, description about CIPN severity has been removed, as explained above.

R1-11) Lines 284-290: It would be worthwhile to mention that “female hormone-related diseases” were associated with PIPN within this paragraph since it also fits with this hypothesis (currently lines 319-322). I think more should be made of this finding in the context of your work.

As suggested by Reviewer #1, comments and discussions concerning the association between the history of female hormone-related diseases and PIPN, citing related papers, have been incorporated into Para 1 in Discussion, (Lines 228-230, 239-246).

R1-12) Lines 314-319: It is confusing that the discussion goes through risk factors, then treatment strategies, and then goes back to risk factors. Please move this content up to before the treatment strategies paragraph.

According to Reviewer #1, description about the association between risk factors and prescription drug use for PIPN has been removed, as explained above. Alternatively, come discussion about the trends and of prescription drug use for treatment of PIPN (Lines 262-271).

R1-13) Table 1: Replace “operation” with “surgery”

As suggested by Reviewer #1, ‘operation’ has been replaced with ‘surgery’ in new Table 2 (Lines 169-170). 

Responses to Reviewer #2

We thank Reviewer #2 for considering that our manuscript is interesting, and for providing important suggestions.

R2-1) I'm not familiar with the herbal medicine goshajinkigan. The article looks at patients prescribed the medicine, but is it possible for patients to be taking it without a prescription (e.g. over-the-counter, given that is herbal). If so, how do you know other patients were not taking it which would skew the results?

This is an important comment that we have to address. Goshajinkigan, a polyherbal medicine that legally requires a prescription to be dispensed in Japan, contains fixed proportions of 10 crude herbal extracts, i.e. achyranthes root, rehmannia root, dioscorea, rhizome, cornus fruit, alisma rhizome, plantago seed moutan bark, pariah sclerotium, aconite root and cinnamon bark (Kono et al., 2013, Cancer Chemother Pharmacol 72, 1283). Goshajinkigan is widely used for treatment of diabetic neuropathy, and appears to have limited effectiveness on CIPN (Kono et al., 2013, Cancer Chemother Pharmacol 72, 1283; Hashino et al., 2018, Int J Clin Oncol 23, 434; Kuriyama et al., 2018, Support Care Cancer 23, 1843). In the present study, we could retrospectively collect information of Goshajinkigan use from medical records. Considering the reviewer’s comment, more explanation of Goshajinkigan has been incorporated into the text (Lines 262-267).

R2-2) Inclusion criteria (line 94) says: no administration of oxaliplatin or vinca alkaloids. Presumably this means "no prior use of" the drugs as there are patients included it the analysis who were given a platinum. Suggest slight rewording to make it clear.

We thank Reviewer #2 for making an important comment. As the reviewer suggests, we have intended to mean “no prior use of”, and modified the text (Lines 101-103).

R2-3) Line 128. The study included only 283 patients, so why does the data in the results (lines 128-129) give numbers for patients also excluded from the study. How is that information relevant? Suggest this is changed so only discusses patients included in the study.

The description of the results in the original manuscript might not be clear. Considering the reviewer’s comment, we have simply described “In this study, 283 patients including 162 breast and 121 gynecologic cancer survivors who met inclusion criteria (S1 Fig) were enrolled for statistical analysis (Table 1)” in the revised manuscript (Lines 134-135).

---

## [Editor Report · Decision Letter 1]

3 Dec 2021

Risk factors and pharmacotherapy for chemotherapy-induced peripheral neuropathy in paclitaxel-treated female cancer survivors: A retrospective study in Japan

PONE-D-21-29568R1

Dear Dr. Kawabata,

We’re pleased to inform you that your manuscript has been judged scientifically suitable for publication and will be formally accepted for publication once it meets all outstanding technical requirements.

Kind regards,

Nial J. Wheate, Ph.D.

Academic Editor

PLOS ONE
---

## [Editor Report · Acceptance letter]

21 Dec 2021

PONE-D-21-29568R1 

Risk factors and pharmacotherapy for chemotherapy-induced peripheral neuropathy in paclitaxel-treated female cancer survivors: A retrospective study in Japan 

Dear Dr. Kawabata:

I'm pleased to inform you that your manuscript has been deemed suitable for publication in PLOS ONE. Congratulations! Your manuscript is now with our production department. 

Kind regards, 

on behalf of

Dr. Nial J. Wheate 

Academic Editor

PLOS ONE